# Enhanced Systemic Humoral Immune Response Induced in Mice by Generalized Modules for Membrane Antigens (GMMA) Is Associated with Affinity Maturation and Isotype Switching

**DOI:** 10.3390/vaccines11071219

**Published:** 2023-07-08

**Authors:** Diego Piccioli, Francesca Buricchi, Marta Bacconi, Nicoletta Bechi, Barbara Galli, Francesca Ferlicca, Enrico Luzzi, Elena Cartocci, Sara Marchi, Giacomo Romagnoli, Renzo Alfini, Roberta Di Benedetto, Simona Gallorini, Silvana Savino, Brunella Brunelli, Erika Bartolini, Francesca Micoli

**Affiliations:** 1GSK Vaccines, 53100 Siena, Italy; 2GSK Vaccines Institute for Global Health (GVGH), 53100 Siena, Italy

**Keywords:** GMMA, antigen carrier, humoral immune response, enhancement, affinity maturation, isotype switching

## Abstract

Generalized Modules for Membrane Antigens (GMMA) are outer membrane vesicles derived from Gram-negative bacteria that can be used to design affordable subunit vaccines. GMMA have been observed to induce a potent humoral immune response in preclinical and clinical studies. In addition, in preclinical studies, it has been found that GMMA can be exploited as optimal antigen carriers for both protein and saccharide antigens, as they are able to promote the enhancement of the antigen-specific humoral immune response when the antigen is overexpressed or chemically conjugated to GMMA. Here we investigated the mechanism of this GMMA carrier effect by immunizing mice and using factor H binding protein and GMMA of *Neisseria meningitidis* B as an antigen–GMMA model. We confirmed that the antigen displayed on the GMMA surface increased the antigen-specific IgG production and, above all, the antibody functionality measured by the serum bactericidal activity. We found that the enhancement of the bactericidal capacity induced by GMMA carrying the antigen on the surface was associated with the increase in antibody affinity to the antigen, and with the switching toward IgG subclasses with more bactericidal potential. Thus, we conclude that the potent carrier effect of GMMA is due to their ability to promote a better quality of humoral immunity.

## 1. Introduction

Generalized Modules for Membrane Antigens (GMMA) is a technology platform suitable for vaccine design [1]. Currently, there are three GMMA-based vaccines in clinical development against shigellosis, invasive nontyphoidal salmonellosis, and gonorrhea (https://clinicaltrials.gov/ct2/show/NCT05073003; https://www.clinicaltrials.gov/ct2/show/NCT05480800 (accessed on 30 May 2023); ISRCTN—ISRCTN51750695: Salmonella vaccine study in Oxford; https://ichgcp.net/clinical-trials-registry/NCT05630859 (accessed on 30 May 2023)). GMMA are outer membrane vesicles (OMVs) blebbing out spontaneously from Gram-negative bacteria and, as such, represent the surface of the source bacterium [1]. Thus, GMMA are subunits of the bacteria from which they derive, presenting surface-exposed antigens in the native cell compartment [1]. In addition, GMMA can be recovered from the culture supernatants of grown bacteria through relatively easy and inexpensive methods of purification, allowing affordable manufacturing of GMMA-based vaccines [1]. Of note, the bacterium used as the source of GMMA can be genetically manipulated to obtain GMMA with specific features useful to design a fit-for-purpose GMMA-based vaccine, for example, by deleting antigens detrimental to the effectiveness or safety of the vaccine or overexpressing critical vaccine antigens to elicit protection [1]. Importantly, the bacterium used to generate GMMA can be manipulated to express antigens derived from different strains or from different pathogens, allowing the opportunity to use GMMA for the design of combination vaccines [2,3]. It has also been observed that antigen overexpression on the GMMA surface can increase the immunogenicity of the antigen itself [2,3]. Thus, GMMA might also be exploited as carriers to design highly effective standalone or combination vaccines.

We expanded this concept by chemically conjugating protein or polysaccharide antigens to GMMA and confirming that the display of antigens on the GMMA surface enhanced the antigen-specific humoral immune response without affecting the antibody response to GMMA used as carriers [4]. In particular, we showed that the antigen-specific antibody production and functionality improved immunizing with GMMA with the antigen attached to the GMMA surface, compared to either the physical mixture of antigen plus GMMA or the antigen alone [4].

In this study, we wanted to shed light on the mode of action of this GMMA carrier effect. We used *Neisseria meningitidis* B (MenB) GMMA not expressing factor H binding protein (fHbp) and fHbp as a model antigen to evaluate the quantity and quality of the antigen-specific humoral immune response after immunization of the mouse animal system. We considered the MenB model, GMMA plus fHbp, particularly interesting as the bactericidal activity in humans is a correlate of protection for MenB vaccines and, consequently, is an optimal readout of vaccine efficacy in preclinical studies [5].

We found that the carrier effect of GMMA was associated with higher antibody affinity for the antigen and the switching toward IgG subclasses with superior bactericidal activity.

## 2. Materials and Methods

### 2.1. Animals and Injections

Animal studies were carried out at the GSK Animal Facility in Siena, Italy, in compliance with the Italian D. Lgs. n. 26/14, the European Directive 2010/63/UE, and the GSK policy on the care, welfare, and treatment of animals. The animal protocol used for these studies was approved by the Animal Welfare Body of GSK Vaccines, Siena, Italy, and the Italian Ministry of Health. C57BL/6 mice were 8-week-old females at day 0 of each study. CD1 mice were 7-week-old females at day 0 of the study. Mice were injected in the calf muscle three times on days 0, 21, and 35. C57BL/6 mice were treated with a total volume of 40 μL per animal, injecting 20 μL per leg. CD1 mice were treated with a total volume of 50 μL per animal, injecting 25 μL per leg. The difference in the volume of the inocula in the study conducted with CD1 mice was due to reasons of concentration of immunization material. Blood samples to separate sera were collected following the schedule described in the text.

### 2.2. Immunogens

GMMA, GMMA overexpressing fHbp variant 3, and fHbp variant 3 were produced and analyzed as previously described [6].

### 2.3. Synthesis and Characterization of the fHbp-GMMA Conjugates

fHbp variant 3 (hereinafter referred to as fHbp) was conjugated to MenB GMMA surface proteins through BS3 chemistry as already described [4,7]. Conjugate formation was verified by SDS-PAGE/Western Blot analysis. The amount of linked fHbp was estimated by aminoacidic analysis, as previously reported [4].

### 2.4. Formulation

Formulations were prepared by adsorbing fHbp protein, GMMA, or GMMA conjugates onto Aluminum Hydroxide (Alum) suspension (3 mg/mL, corresponding to 1 mg/mL Al^3+^). Formulations without Alum were also prepared. When required, resuspended liquid Monophosphoryl Lipid A (MPLA) (Avanti Polar Lipids, Alabaster, AL, USA) was added to the respective formulation. The buffer used for the formulation was histidine pH 6.5. The final concentration of the buffer was fixed at 10 mM, and NaCl was included in each formulation to reach the final osmolality of 308 mOsm/kg. Formulations were prepared following a sequential addition of each component as listed below: (1) water for injection, (2) buffer, (3) Alum or MPLA (when applicable), (4) antigen (when applicable), (5) GMMA (when applicable), and (6) NaCl. In the vaccine formulation, each component was added by gentle pipetting. Before characterization, the formulation was left for 30 min under gentle mixing in a tilting shaker (15 rpm), avoiding vertexing and shaking. Formulations were stored overnight at 4 °C before injection. Formulations passed the testing for pH, osmolality, visible precipitates, endotoxin content, and bioburden; antigen adsorption onto Alum, identity, and integrity were evaluated via reducing SDS-PAGE. Alum-based formulations were centrifuged, and Alum pellets were resuspended in SDS-desorption buffer (sodium phosphate 900 mM, 2% SDS, 6% glycerol, Bromophenol blue 0.06%, and DTT 1.5%), incubated at 95 °C for 10 min and then loaded on gels (Invitrogen NuPAGE pre-cast 4–12% Bis-Tris Gels). Antigen in the supernatant was precipitated using sodium deoxycholate and trichloroacetic acid, centrifuged, resuspended in loading sample buffer (Tris-HCl 0.5 M pH 8, 2% SDS, 6% glycerol, and Bromophenol blue), incubated at 95 °C for 10 min and entirely loaded on gel (Invitrogen NuPAGE, Thermo Fisher Scientific, Waltham, MA, USA). The size distribution of Alum adsorbed (2 μm average size) and non-adsorbed (100 nm average size) GMMA formulations was measured by static and dynamic laser light scattering, respectively.

### 2.5. Enzyme-Linked Immunosorbent Assay (ELISA)

Microtiter plates were coated overnight at 4 °C with 0.015 μM of fHbp (741 2-3-1) or 1.0 μg/mL of fHbp variant 3 (741 Var.3) in PBS. After overnight incubation at 4 °C, plates were washed three times with washing buffer (0.05% Tween-20 in PBS 0.074 M) and saturated with saturation buffer (2.7% polyvinylpyrrolidone (Sigma-Aldrich, Merck, Burlington, MA, USA) in water) for 2 h at 37 °C. Then, plates were washed three times with washing buffer. Sera samples were then serially diluted in the dilution buffer (PBS 0.074 M, 1% Bovine Serum Albumin, and 0.05% Tween20) and incubated for 2 h at 37 °C. After incubation, plates were washed three times with the washing buffer and incubated for 90 min at 37 °C with alkaline phosphatase-conjugated anti-mouse antibodies, diluted 1:2000 in dilution buffer. After incubation, plates were washed three times with the washing buffer and were then incubated at room temperature for 30 min with a p-nitrophenyl phosphate solution. After the incubation, 100 μL/well of NaOH solution 4N was added to stop the enzymatic reaction, and the optical density was analyzed using a plate reader at a dual wavelength of OD 405/620–650 nm. Antibody titers were quantified as the dilution of serum that gives an absorbance of 0.4 OD.

### 2.6. Serum Bactericidal Assay (SBA)

Serum bactericidal antibody activity against MenB strains with mice antisera was evaluated as previously described, with pooled baby rabbit serum at the final concentration of 25% used as the complement source [8]. Bacteria were sub-cultured overnight on chocolate agar plates and resuspended in liquid Mueller–Hinton culture medium. Bacteria were grown in broth containing 0.25% (*w*/*v*) glucose until an optical density of 0.25 at 600 nm. SBA titers were determined as the reciprocal serum dilution that resulted in at least a 50% reduction in colony forming units (CFU) after 60 min of incubation, with respect to the average number of CFU calculated on the experimental controls. Controls were (a) bacteria incubated with active complement and without serum and (b) bacteria incubated with inactive complement and with serum. The MenB strain used in this study was the M01-240320 (UK320) strain, carrying and expressing fHbp variant 3 (3.45 fHbp peptide).

### 2.7. Avidity Index Determination

The Avidity Index (AI) (percentage of high-affinity antibodies on total antigen-specific antibodies) was determined by a previously described avidity ELISA method in which ammonium thiocyanate was used as a chaotropic agent able to disrupt the lower affinity antigen–antibody binding [9]. This method was applied to Gyrolab^®^ system with a 4-step method, provided with an additional wash with ammonium thiocyanate 1.5 M (Sigma-Aldrich, Merck, Burlington, MA, USA) or with PBS, before the addition of the secondary antibody with respect to the classical three-step method (capture antigen–sample–secondary antibody). The detection of fluorescence in Gyrolab^®^ system is based on the emission of Alexa 647 detection antibody when hit by a laser scan which is automatically detected and analyzed by Gyrolab^®^ Evaluator software. The Avidity Index was calculated as the fluorescence intensity (FI) measured after ammonium thiocyanate 1.5 M wash/FI after PBS wash × 100. All samples were tested at 1:200 dilution in Rexxip H buffer (Gyros Protein Technologies AB, Uppsala, Sweden). As capture reagent, biotinylated fHbp variant (fHbp-v3) was used at 100 µg/mL (diluted in PBS-Tween20 0.01%). The antigen was biotinylated using EZ-Link^®^ sulfo-NHS-LC-Biotin (Thermo Fisher Scientific, Waltham, MA, USA) at a molar excess of 10 moles of biotin:1 mole of protein. As detection reagent goat anti-human IgG, Fcγ fragment specific-Alexa 647 (Jackson ImmunoResearch, Ely, UK) was used at 25 nM. Dilution was performed in Rexxip F^TM^ buffer (Gyros Protein Technologies AB, Uppsala, Sweden).

### 2.8. IgG Isotype Distribution Analysis

IgG isotype-specific antibody titers were determined at Gyrolab^®^, using a three-step method with 100 µg/mL biotinylated fHbp-v3 as capture reagent, sera diluted 1:200 as samples and 25 nM A647-secondary antibodies specific for each IgG isotype as detection reagent (sheep anti-human IgG1/2/3/4 from Binding Site). The secondary antibodies were labeled with Alexa 647 at a molar excess of 10 moles of dye:1 mole of antibody using the Alexa Fluor^®^ monoclonal antibody labeling kit from Molecular Probes (Invitrogen, Thermo Fisher Scientific, Waltham, MA, USA). Unlabeled dye was removed using Zeba desalting spin column (Thermo Fisher Scientific, Waltham, MA, USA), following manufacturer’s instructions. The IgG isotype titers were calculated as FI and can be compared between the same isotype only.

### 2.9. Statistical Analysis

The animal studies were exploratory, and no statistical success criteria were pre-defined. Consequently, the sample size was not computed to ensure a target power. Based on past experience, we immunized 10 animals per group, as this number generally allows us to reach meaningful results with the same species. We applied a post hoc statistical analysis by using GraphPad and comparing couples of study groups with the two-tailed Mann–Whitney test.

## 3. Results

First, we wanted to confirm the carrier effect of GMMA by using the MenB model.

Previously we have shown that CD1 mice immunized subcutaneously twice with MenB fHbp generated a higher fHbp specific antibody response when fHbp was chemically conjugated to *Salmonella* Typhimurium GMMA, compared to fHbp alone or physically mixed with *Salmonella* GMMA [4]. In particular, we showed that the serum fHbp-specific antibody production was superior after the first immunization, but not after the second one when immunizing with fHbp conjugated to *Salmonella* GMMA, compared to the fHbp and GMMA physical mixture [4]. Despite that, the sera collected after the second immunization with fHbp conjugated to *Salmonella* GMMA demonstrated superior bactericidal activity against MenB, compared to the sera of mice immunized with fHbp physically mixed with GMMA [4]. In addition, the sera collected after the second immunization with fHbp-GMMA conjugate were the only ones showing bactericidal activity also against MenA and MenW strains, indicating not only a more potent functionality, but also a potentially improved breadth of the humoral immune response when immunizing with the conjugate [4].

Here, in our first study, animals of a different mouse strain (C57BL/6) were immunized intramuscularly three times, and sera were collected two weeks after the last injection. Mice were immunized with fHbp alone at three different doses (0.1, 1, and 10 μg), with fHbp at a 0.1 μg dose physically mixed with MenB GMMA not expressing fHbp (1.9 μg as protein content) or with fHbp chemically conjugated to the surface of MenB GMMA (fHbp-GMMA conjugate), in order to have a total amount of 2 μg of fHbp-GMMA conjugate containing about 0.1 μg of fHbp and 1.9 μg of MenB GMMA. Thus, the physical mixture and the conjugate had the same content of fHbp and GMMA. All immunizations were executed with immunogens adsorbed on Alum.

As reported in Figure 1, the display of the fHbp antigen on the MenB GMMA surface generated a systemic antigen-specific humoral immune response which was substantially enhanced in quantity and quality. In fact, at the antigen dose of 0.1 μg, the immunization with the GMMA-conjugate led to a serum antigen-specific total IgG production that was significantly higher compared to immunizations with antigen alone or physically mixed with GMMA (Figure 1A). The addition of GMMA in a physical mixture with the antigen did not induce any enhancement of the antigen-specific IgG response (Figure 1A). Immunizations with 1 or 10 μg doses of the fHbp antigen increased the antigen-specific IgG response compared to 0.1 μg dose, but interestingly, the GMMA-fHbp conjugate promoted an antibody response against the antigen that was still much higher compared to the immunization with 1 μg dose of antigen alone, and was comparable to the antigen alone at 10 μg dose (Figure 1A). Thus, the presence of fHbp on the GMMA surface resulted in a serum antigen-specific antibody response after immunization which could be obtained with the antigen alone when using a 100-times greater amount of it.

The functionality of the antibody response was evaluated by measuring the bactericidal activity of the animal sera against a reference strain for the fHbp antigen. The killing activity against this bacterial strain was mediated only by fHbp and not by the GMMA component. Interestingly, immunizing with 0.1 μg of antigen, the bactericidal activity was detectable only with sera of mice immunized with the GMMA conjugates (Figure 1B). More importantly, the serum bactericidal activity obtained from immunizing with the GMMA conjugate was superior to those observed after immunization with a 10- or 100-times higher amount of fHbp alone (Figure 1B). Thus, comparing mice immunized with 2 μg of fHbp-GMMA conjugate to those immunized with 10 μg of fHbp, although the fHbp-specific serum IgG was similar in the two immunization groups, the functionality of the antibody response was superior in animals immunized with the GMMA conjugate.

In conclusion, the presence of the antigen on the GMMA surface induced, after immunization, an improved antigen-specific systemic humoral immune response, determined by the significant enhancement of the serum IgG antibody production and, above all, the remarkably increased functionality of the serum antibody response. Thus, we demonstrated the potent carrier effect of GMMA also for the MenB model of antigen and GMMA used here.

Although the carrier effect of GMMA is based on the presentation of the antigen on the vesicle surface, we have previously shown that GMMA itself may have an immunostimulatory effect toward physically mixed antigens [4]. Indeed, this was expected, as GMMA contain Toll-like Receptor (TLR) 2 and 4 agonists, such as lipoproteins or peptidoglycan for TLR2 and lipopolysaccharide (LPS) for TLR4, which are well-known stimulatory pathways for immune cell responses [10,11,12,13]. The monophosphoryl lipid A (MPL), a TLR4 agonist, possesses a well-described adjuvant activity, and it is utilized as a component of adjuvants contained in licensed vaccines [14] (https://www.shingrix.com/; https://www.who.int/news/item/06-10-2021-who-recommends-groundbreaking-malaria-vaccine-for-children-at-risk (accessed on 30 May 2023)).

Thus, in order to more deeply understand the importance of the carrier effect of GMMA, we better explored the adjuvant properties of GMMA themselves compared to GMMA used as carriers. To this aim, we immunized mice with fHbp alone or physically mixed with different doses of GMMA. In more detail, we used the same mouse strain (C57BL/6), immunization schedule (three injections), and administration route (intramuscular) of the precedent study reported in Figure 1. However, in this second study, we used 1 μg of fHbp, as previously, at this antigen dosage and in formulation with Alum, we observed an immunogenic effect, although low, while at 0.1 μg antigen dose, we found a very limited antibody response and no antibody functionality (Figure 1). GMMA were used at 1.9, 19, and 100 μg doses, as 1.9 μg was the dose used in the previous study to compare the physical mixture with the conjugate, and it is used here as the starting point of the dose-escalation study (10-fold step). The fHbp and GMMA physical mixtures were injected in the absence of Alum to better evaluate the adjuvant effect of GMMA. As a control, we immunized mice either with fHbp alone or adjuvanted with Alum or MPL. The fHbp-GMMA conjugates used as comparators were injected, not adsorbed on Alum, at 2 μg dose and contained about 0.1 μg of fHbp (10 times lower amount of antigen than the other immunizations) and 1.9 μg of GMMA, as in the previous study. Sera of animals were collected after the third immunization.

As reported in Figure 2, GMMA physically mixed with the antigen displayed an adjuvant effect, leading to an enhancement of both antigen-specific antibody production and functionality. Indeed, fHbp alone at 1 μg dose was not able to elicit any humoral response (Figure 2). The immunostimulatory property of GMMA was similar in increasing the dosage and was not significantly different from those of Alum for antibody production (although a trend of increase was observed with Alum, but with high variability) (Figure 2A) and bactericidal activity of sera (Figure 2B). Interestingly, when immunizing with fHbp-GMMA conjugate containing 10 times lower amount of antigen (0.1 μg), both fHbp-specific IgG production and serum bactericidal activity were much higher compared to those of any GMMA physical mixtures, whatever the dosage of GMMA used, even when it was 10 (19 μg) or 50 (100 μg) times higher than those of the conjugate (Figure 2).

In conclusion, we demonstrated that GMMA had an adjuvant potential to elicit a systemic antigen-specific humoral immune response, which was very limited if compared to its carrier effect, confirming that the display on the GMMA surface promotes an optimal humoral immune response against an antigen.

Next, we asked whether immunization with the antigen displayed on the GMMA surface could result in a quality of the antigen-specific antibody response, which could justify the improved functionality of the humoral immune response. Antibodies that recognize an antigen with higher affinity can, in principle, lead to a more potent bacterial killing. Also, it was previously observed that different monoclonal antibodies, recognizing the same epitope of MenB Porin A antigen with similar affinities, demonstrated a substantial difference in bactericidal activity against MenB, showing the following hierarchy: IgG3 >> IgG2b > IgG2a >> IgG1 [15]. With the aim of associating the superior functionality to a better quality of the IgG response, we asked whether the immunization with fHbp displayed on the GMMA surface could promote the production of IgG with higher antigen affinity and an increased generation of antigen-specific IgG subclasses with more potent bactericidal potential, compared to the physical mixture of fHbp and GMMA. 

For this new study, we used CD1 mice injected intramuscularly to confirm previous data, also in outbred mice, where more variability of antibody responses might be expected. We immunized mice three times with MenB GMMA either overexpressing or bearing chemically conjugated fHbp on the surface and with GMMA physically mixed with fHbp, at the same doses for fHbp and GMMA, either in the presence or absence of Alum. Mouse sera were collected the day before each injection and two weeks after the third injection.

We first evaluated the carrier effect.

As shown in Figure 3 and Appendix A, we confirmed that fHbp chemically conjugated or overexpressed on GMMA resulted in a superior systemic humoral immune response compared to immunization with fHbp + GMMA physical mixture. In detail, we observed a higher fHbp-specific serum total IgG production after the first and the second injection, immunizing with fHbp-GMMA conjugate or GMMA overexpressing fHbp compared to fHbp + GMMA physical mixture, while the antigen-specific IgG production became comparable after the third injection (Figure 3A and Appendix A). Despite the similarity of the antigen-specific serum IgG production between mice immunized with GMMA bearing fHbp on the surface and fHbp + GMMA physical mixture after the third injection, the bactericidal activity resulted in much higher immunizing with the fHbp-GMMA conjugate or GMMA overexpressing fHbp (Figure 3B and Appendix A). In addition, the bactericidal activity using sera collected after the second and, above all, the first injection was detectable only when immunizing with the fHbp-GMMA conjugate or GMMA overexpressing fHbp, and not with the fHbp + GMMA physical mixture (Figure 3B and Appendix A). The formulation with Alum showed an added value for antibody production only to the conjugate, but not to GMMA overexpressing fHbp (statistical analysis not shown). However, the functionality of the humoral immune response when immunizing with the conjugate or the GMMA overexpressing fHbp was not affected, either negatively or positively, by using Alum (Figure 3 and Appendix A). Thus, the antigen display on the GMMA surface, with or without formulation with Alum, induced a superior functionality of the humoral immune response after repeated injections compared to antigen physically mixed with GMMA and allowed the generation of a functional humoral immune response after the first or second injection. In conclusion, the carrier effect of GMMA led to a more effective antigen-specific systemic humoral immune response, which occurs already at early time points after the immunization.

As the fHbp-specific serum’s total IgG production was similar to whatever fHbp was present on the GMMA surface or physically mixed with GMMA after three injections, we wanted to understand why the functionality of the sera containing the same content of fHbp-specific IgG was so different at promoting an fHbp-mediated bactericidal activity. We asked whether the quality of the antigen-specific IgG could explain this important difference observed in the functionality. To this aim, we decided to measure the affinity of total IgG for the antigen fHbp and the production of the fHbp-specific IgG subclasses using sera collected after the third injection.

The bactericidal activity was measured on individual mouse sera, and we confirmed that immunization with fHbp-GMMA conjugate or GMMA overexpressing fHbp induced significantly higher bactericidal titers compared to the fHbp + GMMA physical mixture (Figure 4A and Appendix A). Then, we found that the affinity of the serum total IgG to fHbp was superior when mice were immunized with the fHbp-GMMA conjugate or GMMA overexpressing fHbp compared to the fHbp + GMMA physical mixture (Figure 4B and Appendix A). Analyzing the serum IgG subclasses, we observed that the production of IgG3, IgG2a, or IgG2b was substantially increased after immunization with the fHbp-GMMA conjugate or GMMA overexpressing fHbp compared to the fHbp + GMMA physical mixture, whereas IgG1 was reduced (in the absence of Alum) or did not change (in the formulation with Alum) (Figure 4C,D and Appendix A). When immunizing with GMMA overexpressing fHbp, a more prominent difference in IgG2a, IgG2b, and IgG3 production between GMMA bearing fHbp and physical mixtures was measured immunizing with Alum (Appendix A). Consistently to what was previously published using monoclonal antibodies against MenB Porin A regarding the killing of MenB, fHbp-specific antisera containing the same amount of total IgG but a higher amount of IgG3 and IgG2 generated a superior bacterial killing. Thus, by using our MenB model, the immunization with an antigen displayed on the GMMA surface resulted in a systemic antigen-specific humoral immune response with improved functionality, which was associated with higher antigen affinity of IgG and an increased generation of IgG subclasses with superior bactericidal potential.

We concluded that the carrier effect of GMMA for the antigen was associated with affinity maturation and isotype switching of antibodies.

## 4. Discussion

The major finding of our work drives the further investigation of the mode of action of the GMMA carrier effect from different perspectives.

First, we believe that, as a follow up to our study, it would be of critical importance to analyze the antigen-specific T follicular helper (Tfh) cell response upon immunization with GMMA bearing an antigen on the surface, as this cell population is primarily involved in inducing affinity maturation within the germinal center [16,17,18,19]. In addition to that, the role of monocyte-derived dendritic cells should also be investigated, as these cells have been described to play a key role in promoting the Tfh cell response [20].

Secondly, it would be critical as well to study the trafficking of GMMA vesicles within the lymphoid organs and the role of subcapsular sinus macrophages (SCSMs) and/or follicular dendritic cells (FDCs) in the GMMA carrier effect. SCSMs are key transporters of antigens (especially particulated ones like GMMA) to B cells and can drive extrafollicular B cell activation, where isotype switching can take place [19,21,22,23]. FDCs play a critical role in promoting germinal center formation and can also participate in inducing isotype switching [24,25].

From this last perspective, the results presented here are particularly intriguing because the antigen presentation by FDCs to cognate B cells has been demonstrated as pivotal in promoting an anti-GMMA systemic humoral immune response by using *Shigella* and *Salmonella* GMMA as models [26]. Thus, it would certainly be a priority to confirm whether the role of FDCs is also critical for MenB GMMA immunogenicity. Together with Tfh cells, FDCs represent the most important cell type that promotes B cell responses; thus, their involvement in the mode of action of the GMMA carrier effect should be a critical next step of investigation [16,18,25].

## 5. Conclusions

This work represents a significant advancement in the knowledge of the GMMA technology platform. We discovered that the improvement in the humoral immune response observed when an antigen is displayed on the GMMA surface was due to the induction of a better quality of the antigen-specific antibody response. In particular, we found that the affinity of specific IgG for the antigen was significantly enhanced and that the pattern of antigen-specific IgG subclasses was substantially changed. In the MenB GMMA-antigen model used in this work, we observed a switch toward a significant increase in antigen-specific IgG3 and IgG2 subclasses, which was consistent with the improved functionality of the antibody response, measured as the bactericidal capacity of immune sera. Thus, we concluded that the potent immunogenic effect of GMMA as antigen carriers is associated with affinity maturation and isotype switching. Several immunological mechanisms can underlie this phenomenon, and they should be addressed in the coming future. We believe indeed that these mechanisms might be leveraged to generate more immunogenic and less reactogenic GMMA and ultimately to design more effective GMMA-based vaccines in the future.

## Figures and Tables

**Figure 1 vaccines-11-01219-f001:**
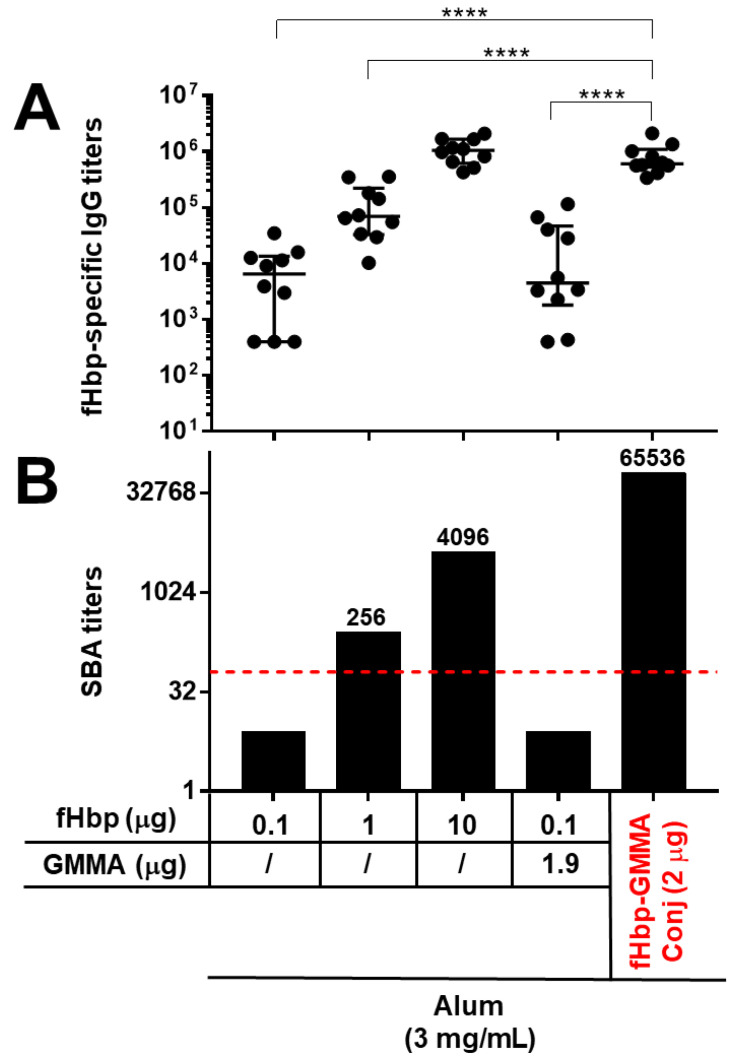
The carrier effect of GMMA: immunization with GMMA bearing an antigen chemically attached to the surface resulted in a substantial enhancement of the antigen-specific humoral immune response. C57BL/6 mice were immunized intramuscularly three times with fHbp alone, physically mixed with MenB GMMA (fHbp + GMMA), or chemically conjugated to MenB GMMA (fHbp-GMMA Conj, in red). MenB GMMA were deleted by fHbp. The dosage of fHbp, GMMA, and conjugates are reported. The dose of conjugates (2 μg) contained the same amount of fHbp (0.1 μg) and GMMA (1.9 μg) as in the physical mixture. All immunogens were adsorbed on Alum used at 3 mg/mL, as indicated. Sera were collected two weeks after the last immunization and analyzed by ELISA and SBA assays. (**A**) fHbp-specific total IgG titers measured in individual mouse sera (black dots). Data are reported using base-10 logarithmic scale (*y*-axis). (**B**) SBA titers against UK320 MenB strain (reference for fHbp variant 3 (used in this study) in the killing assay) measured in pooled mouse sera from each group of the study. Data are reported using base-2 logarithmic scale (*y*-axis). SBA titers are indicated over the columns. The red dotted line set at the SBA titer of 64 represents the threshold of significance for the assay to unequivocally establish a killing activity. An increase of at least 8-fold in SBA titer comparing two serum pools is considered biologically significant. IgG ELISA titers were analyzed by GraphPad applying the two-tailed non-parametric Mann–Whitney test. The titers are graphically represented with median and interquartile ranges. **** *p*-value: <0.0001.

**Figure 2 vaccines-11-01219-f002:**
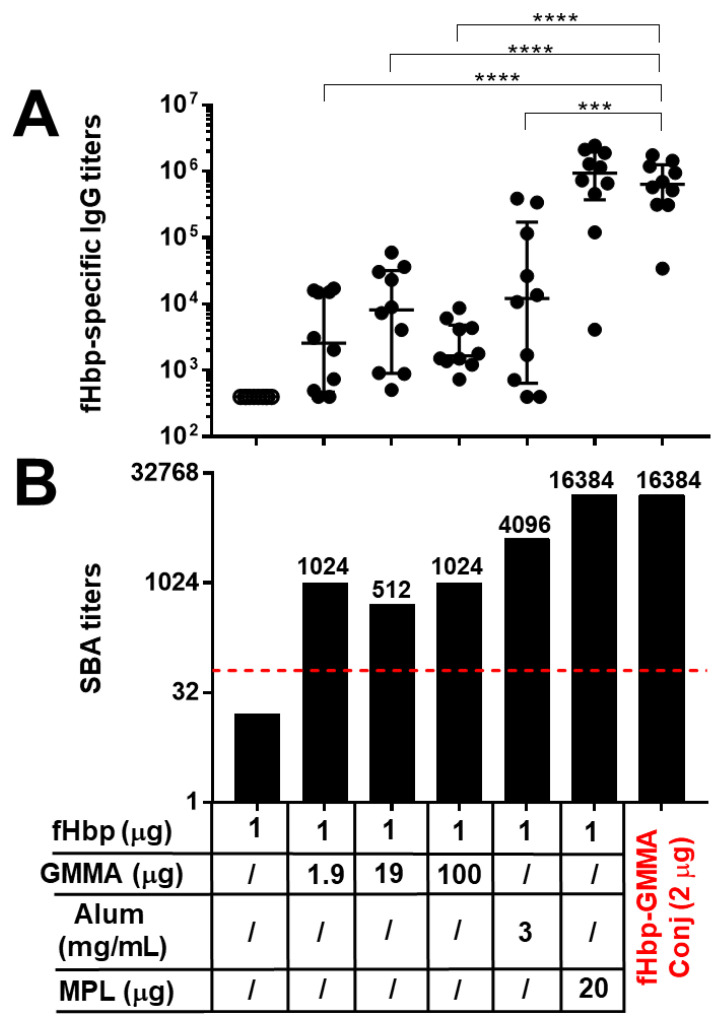
GMMA showed an adjuvant effect that was substantially lower compared to the carrier effect. C57BL/6 mice were immunized intramuscularly three times with fHbp alone, physically mixed with different doses of MenB GMMA, physically mixed with MPL, adsorbed on Alum, or chemically conjugated to MenB GMMA (fHbp-GMMA Conj, in red). MenB GMMA were deleted by fHbp. The dosage of fHbp and GMMA are reported. Alum was used at 3 mg/mL, whereas the dosage of MPL was 20 μg, as indicated. The dose of conjugates (2 μg) contained 0.1 mg of fHbp and 1.9 mg of GMMA. Sera were collected two weeks after the last immunization and analyzed by ELISA and SBA assays. (**A**) fHbp-specific total IgG titers measured in individual mouse sera (black dots). Data are reported using base-10 logarithmic scale (*y*-axis). (**B**) SBA titers against UK320 MenB strain (reference for fHbp variant 3 (used in this study) in the killing assay) measured in pooled mouse sera from each group of the study. Data are reported using base-2 logarithmic scale (*y*-axis). SBA titers are indicated over the columns. The red dotted line set at the SBA titer of 64 represents the threshold of significance for the assay, to unequivocally establish a killing activity. An increase of at least 8-fold in SBA titer comparing two serum pools is considered biologically significant. IgG ELISA titers were analyzed by GraphPad, applying the two-tailed non-parametric Mann–Whitney test. The titers are graphically represented with median and interquartile ranges. **** *p*-value: <0.0001. *** *p*-value: <0.001.

**Figure 3 vaccines-11-01219-f003:**
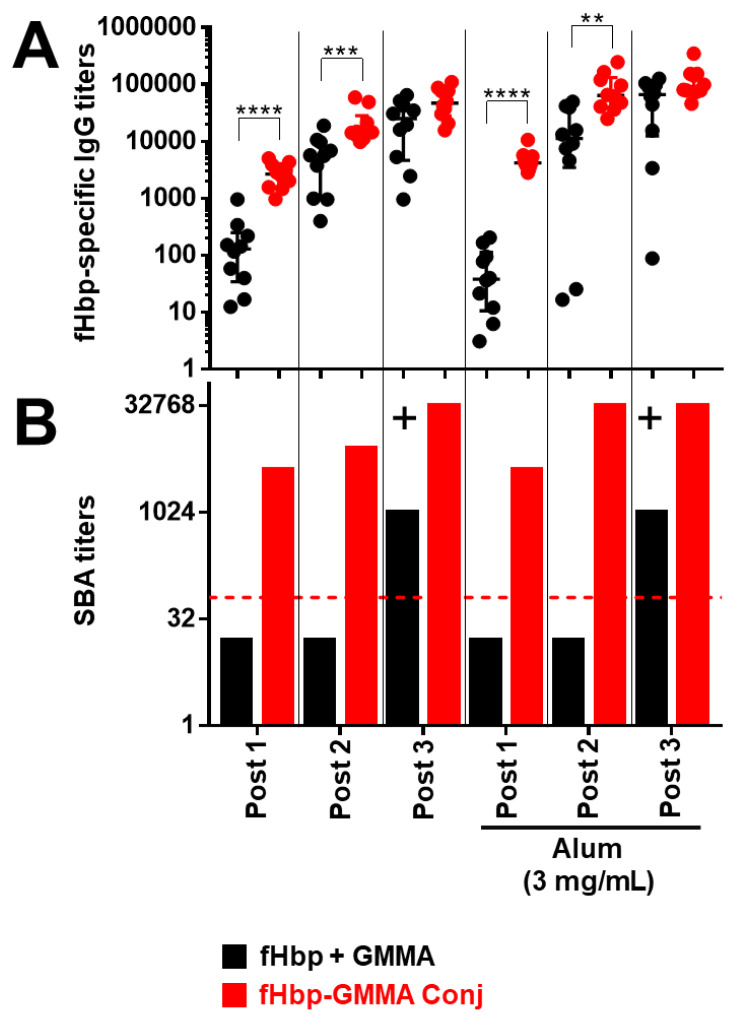
GMMA carrier effect along the different time points of the immunization schedule. CD1 mice were immunized intramuscularly three times with fHbp physically mixed (fHbp + GMMA; black dots or columns) or chemically conjugated (fHbp-GMMA Conj; red dots or columns) to MenB GMMA, either adsorbed or not on Alum, at the same doses of fHbp and GMMA. MenB GMMA were deleted by fHbp. The dosages of fHbp and GMMA (as protein content) were 0.1 μg and 0.6 μg, respectively. The total dose of conjugates was 0.7 μg. Alum was used at 3 mg/mL, as indicated. Sera were collected the day before the first, the second (Post 1), and the third (Post 2) immunization and two weeks after the third (Post 3) immunization. Mouse sera were analyzed by ELISA and SBA assays. (**A**) fHbp-specific total IgG titers measured in individual mouse sera (dots) Post 1, Post 2, and Post 3. Data are reported as base-10 logarithmic scale (*y*-axis). (**B**) SBA titers against UK320 MenB strain (reference for fHbp variant 3 (used in this study) in the killing assay) measured in Post 1, Post 2, and Post 3 pooled mouse sera from each group of the study. Data are reported as base-2 logarithmic scale (*y*-axis). The red dotted line set at the SBA titer of 64 represents the threshold of significance for the assay, to unequivocally establish a killing activity. An increase of at least 8-fold (+) in SBA titer comparing two serum pools is considered biologically significant. IgG ELISA titers were analyzed by GraphPad, applying the two-tailed non-parametric Mann–Whitney test. The titers are graphically represented with median and interquartile ranges. **** *p*-value: <0.0001. *** *p*-value: <0.001. ** *p*-value: <0.01.

**Figure 4 vaccines-11-01219-f004:**
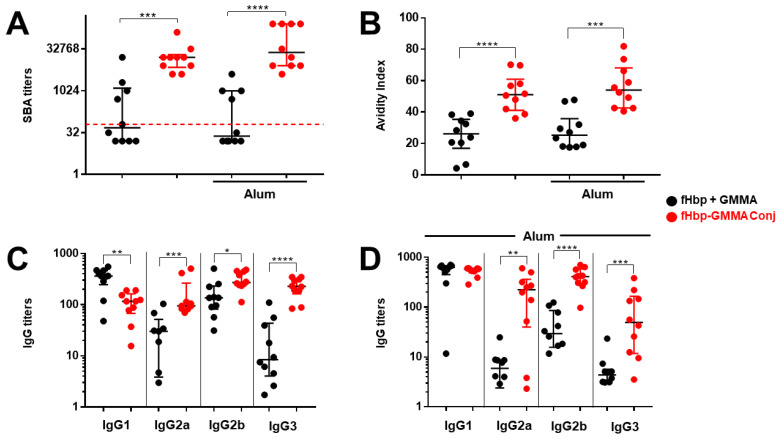
The superior functionality of the humoral immune response induced immunizing with GMMA bearing an antigen on the vesicle surface was associated with affinity maturation and isotype switching. Same experiment described in Figure 3. Analysis of Post 3 sera of mice immunized with fHbp + GMMA physical mixture (black dots) or fHbp-GMMA conjugates (red dots), with and without Alum, as indicated. (**A**) SBA titers against UK320 MenB strain (reference for fHbp variant 3 (used in this study) in the killing assay) measured in individual mouse sera. Data are reported as base-2 logarithmic scale (*y*-axis). (**B**) Avidity Index of fHbp-specific IgG, measured in individual mouse sera. Data are reported as linear scale (*y*-axis). (**C**,**D**) Titers of fHbp-specific IgG subclasses (IgG1, IgG2a, IgG2b, and IgG3) measured in individual mouse sera immunized in (**C**) absence or (**D**) presence of Alum. Data are reported as base-10 logarithmic scale (*y*-axis). The red dotted line set at the SBA titer of 64 represents the threshold of significance for the assay, to unequivocally establish a killing activity. IgG ELISA titers were analyzed by GraphPad, applying the two-tailed non-parametric Mann–Whitney test. The titers are graphically represented with median and interquartile ranges. **** *p*-value: <0.0001. *** *p*-value: <0.001. ** *p*-value: <0.01. * *p*-value: <0.05.

## Data Availability

The data presented in this study are available on request from the corresponding author. The data are not publicly available due to GSK policy for the protection of intellectual property.

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
