# Peer review of "Enhanced Systemic Humoral Immune Response Induced in Mice by Generalized Modules for Membrane Antigens (GMMA) Is Associated with Affinity Maturation and Isotype Switching"

_vaccines, 2023, doi:10.3390/vaccines11071219_

Round 1

Reviewer 1 Report

The manuscript by Piccioli et la describes the mechanism of action of Generalised Modules for Membrane Antigens (GMMA),  used as vaccine carrier in the mouse model. In particular,  the authors used the Neisseria meningitidis B (MenB) GMMA not expressing factor H binding protein (fHbp) or the soluble fHbp factor, or the fHbp chemically conjugated on the GMMA surface  to evaluate the quantity and quality of the antigen-specific humoral immune response after immunization of mice. The in vivo data have been confirmed into two different mice models, thus reinforcing the value of the results.

The work is well described, and the results are presented in a clear way. One of the major findings about the quality of the response observed is related to the IgG subclass; the author should better discuss what is known about the different the bactericidal activity of the different IgG subclasses

The authors have combined the Results and Discussion sections into a single paragraph labeled as Results. This decision has resulted in a lengthy presentation of the results, although they have been adequately discussed. To improve clarity, the authors should consider adhering to the journal guidelines by separating the Results and Discussion sections. Alternatively, they could expand the Conclusions section, which is currently empty, to highlight the main findings of the study.

Minor comments:

Abstract

Introduce the GMMA acronym

Line 19, correct with “displayed “

 Materials and Methods

The immunization schedule should be reported in paragraph “2.1. Animals and injections” and not only in the text

Please justify the use of two different dosages of inocula for C57BL/6 and CD1 mice

Please explain the specificity of fHbp variant 3, respect to fHbp

Please introduce MPLA (line 91)

Please specific what “T60” means, and possibly rephrase the sentence “in control samples: Complement Dipendent Control and Complement Independent Control.”

In the Avidity Index assay and IgG isotype distribution analysis., please specific how you measure the Fluorescence in the assay

Please indicate what means “post 3” in line 157

Author Response

We thank the reviewer for appreciating our work and the valuable comments that improved our manuscript.

We modified the manuscript according to the reviewer's suggestions. Regarding the comment on "post 3", we cancelled this word as it is indeed useless and misleading for the description of the methodology.

The reviewer will find enclosed the modified manuscript containing also additional changes made to follow comments of other reviewers.

We believe that the manuscript is now ready for publication.

Reviewer 2 Report

I indicated above that the manuscript would benefit from minor editing.  This is because I found many sentences to be rather long.  This is perhaps more of a stylistic preference but I do believe that the paper would be easier to read if it were modified with an eye towards making every sentence as succinct/tight as possible.

misspelled words on lines 134 and 143

Author Response

We thank the reviewer for appreciating our work.

We modified the line 134 that was unclear.

The reviewer will find enclosed the modified manuscript containing also additional changes made to follow comments of other reviewers.

We believe that the manuscript is now ready for publication.

Reviewer 3 Report

In this paper, the authors investigated the mechanism of the GMMA carrier effect by immunizing mice and using factor H binding protein and GMMA of Neisseria meningitidis B as antigen-GMMA model. The authors provided solid evidence indicating that the enhancement of the bactericidal capacity induced by GMMA carrying the antigen on the surface was associated with increased antibody affinity to the antigen and switching toward IgG subclasses with more bactericidal potential. Thus, we conclude that the potent carrier effect of GMMA is due to its ability to promote a better quality of humoral immunity. I only have several minor comments listed below.

1. The discussion section needs to be marked. 

2. Although the conclusion section is optional, moving the conclusion described on page 5 to the conclusion section will be better. It will be easier for the reader to see the conclusion.

3. In Figure 1, the authors did not describe the dose of conjugate. It is described in Figure 2.

4. In Figure 1B, the y-axis is non-linear. It will be better to use the broken lines to indicate that the scale of the y-axis is not linear. The same suggestion applies to Figures 2B, 3B, 4A, S1B, and S2A.

5. In Figure 2 legend, the units of MPL and conjugates are incorrect. 

6. In all the figures, it is unclear whether each data point of SBA titers from pooled mouse sera is from a single experiment or the average of duplicates.  

It is acceptable.

Author Response

We thank the reviewer for appreciating our work and for the comments that improved our manuscript.

Point-by-point response:

1, 2, 3, 5, 6: Manuscript modified accordingly.

4: The data are represented in Log2 scale. We specified it in the legends, as it was not clear.

The reviewer will find enclosed the modified manuscript containing also additional changes made to follow comments of other reviewers.

We believe that the manuscript is now ready for publication.
